# An in vitro study in separating tensile loads during maxillo-mandibular fixation using wire and/or elastics

**Sudeep Pawa** 📷**, Surakit Visuttiwattanakorn** 📷*

Department of Oral and Maxillofacial Surgery, Faculty of Dentistry, Mahidol University, Bangkok, Thailand

* surakit_vis@mahidol.ac.th

**Data Availability Statement:** All relevant data are within the manuscript and its Supporting Information files.

## Abstract

Intermaxillary fixation (IMF) or maxillo-mandibular fixation (MMF) is a fundamental process in stabilizing the maxilla and mandible through dental maximum intercuspation (MIP) during the management of trauma, orthognathic surgery, and reconstruction. Despite the availability of several techniques in achieving adequate maxillo-mandibular fixation, concerns have been raised regarding the sufficiency of using only latex elastics to counter displacing forces during reduction and fixation. To address this, an in vitro study was conducted to compare the efficacy of three maxillo-mandibular fixation methods: wire, elastics, and a combination of both. Custom-made models simulating dental arches were used, and a vertical separation of up to 1 mm was applied at a loading rate of 1 mm/minute using an Instron 5566 Universal Testing Machine. Tensile loads were recorded at 0.5 and 1 mm vertical separation, with each process repeated 10 times in each sample group. The average peak tensile load was then calculated. Statistical analysis using one-way ANOVA at a significance level of $p < 0.05$ revealed significant differences between all three subject groups. The outcomes of this in vitro study suggest that the combination technique (using both wire and elastics) outperformed the individual methods in achieving robust maxillo-mandibular fixation. This positions the combination technique as the most effective among the evaluated fixation methods.

## 1. Introduction

Intermaxillary fixation (IMF) or maxillo-mandibular fixation (MMF) is a crucial technique that aims for the stabilization of the maxilla and mandible through the achievement of dental maximum intercuspation, particularly in cases of facial trauma, reconstruction, and orthognathic surgery [1]. For example, during facial fractures involving the maxilla or mandible, maxillo-mandibular fixation (MMF) is used to adequately reduce fractured segments prior to internal fixation with plates and screws. Similarly, during orthognathic surgery, surgeons apply maxillo-mandibular fixation at the intermediate stage of bimaxillary surgery or before applying internal fixation [2, 3]. Historically, traditional methods for achieving maxillo-mandibular fixation in dentate patients encompassed techniques like Erich's arch bars or

**Funding:** The author(s) received no specific funding for this work.

**Competing interests:** The authors have declared that no competing interests exist.

interdental wiring [3, 4]. The most recent introduction of intermaxillary fixation screws in 1989 represented a notable advancement in this field [3, 4]. Maxillomandibular screws and orthodontic brackets with surgical hooks have become the two most commonly used techniques in orthognathic surgery [1, 5, 6]. In the past, maxillo-mandibular fixation was performed using wire ligatures, usually achieved with 26 or 24-gauge pre-stretched stainless-steel wires which often caused discomfort, pain, or soft tissue damage for the patients during the immobilization period [1]. However, due to concerns about the potential source of needlestick injuries in the contaminated environment of the oral cavity which represents a health risk for surgeons and assistants, it is no longer considered the gold standard [2].

In response, orthodontic latex elastics are used as an alternative to wire ligatures [7]. A recent study showed that heavy orthodontic elastics had a force decay of 31% after 24 hours in simulated oral environments [6]. Statistical analysis showed statistical significance in force reduction in the first 24 hours however, force decay between 1–14 days was not significant [6]. Evidence suggests elastics offer adequate traction even in scenarios where the use of wires may be limited [2, 6–8]. The goal of maxillo-mandibular fixation is to resist displacing forces whilst the new position of the alveolus or bony segments is established and fixed with internal fixation [2, 9]. Despite the importance of these techniques, a significant research gap exists. Inadequate intermaxillary fixation (IMF) following oral and maxillofacial procedures can lead to malocclusion, delayed healing, bone nonunion or malunion, infections, nerve damage, aesthetic issues, functional difficulties, and chronic pain. Proper alignment and immobilization are crucial to prevent these complications and ensure successful recovery and outcomes. Although, multiple methods for achieving optimal maxillo-mandibular fixation have been proposed and used, no prior lab or clinical investigation has comprehensively examined the tensile loads applied during the application of separating forces in various maxillo-mandibular fixation techniques. The primary objective of this study is to determine which maxillo-mandibular fixation method between wire, elastics, and combination represents the most effective form of maxillo-mandibular fixation method. By providing numerical evidence of distinct tensile loads associated with different maxillo-mandibular fixation techniques, this study aims to address this critical knowledge gap and contribute to the advancement of clinical practices in this domain.

## 2. Materials and methods

Two models as depicted in Fig 1 were designed and fabricated using Siemens NX CAD using computer-aided design and manufacturing with stainless steel. The models were meticulously designed and fabricated to closely replicate both the maxillary and mandibular arches, consisting of seven screws representing seven teeth in each quadrant. Custom-milled screws as illustrated in Fig 2 were fabricated using a wire cut technique to mimic maxillo-mandibular screws for the fixation of ligature wire (Stainless steel 0.020" Gauge 24, Lot No. 14662939, Fort Wayne Metals, Indiana, USA) and orthodontic latex elastics (Ram Zoo Pak Elastics, 6 Oz/170g, ¼", 6.35 mm, Lot No. 082091864, Ormco Corporation, California, USA). This experimental study allowed for a detailed investigation into the mechanics and outcomes of various fixation methods, shedding light on the performance characteristics of wire ligatures and orthodontic latex elastics in the context of maxillo-mandibular fixation.

In order to ascertain the optimal fixation technique, a preliminary pilot study was carried out. This preliminary investigation aimed to identify the most stable fixation pattern among various fixation patterns, as depicted in Appendix A (S1 Fig). Upon analyzing the outcomes of the pilot study, the interlocking with an overlapping elastic group (E1) and the overlapping box wire group (W2) showed the highest ability to resist displacing forces when tensile loads

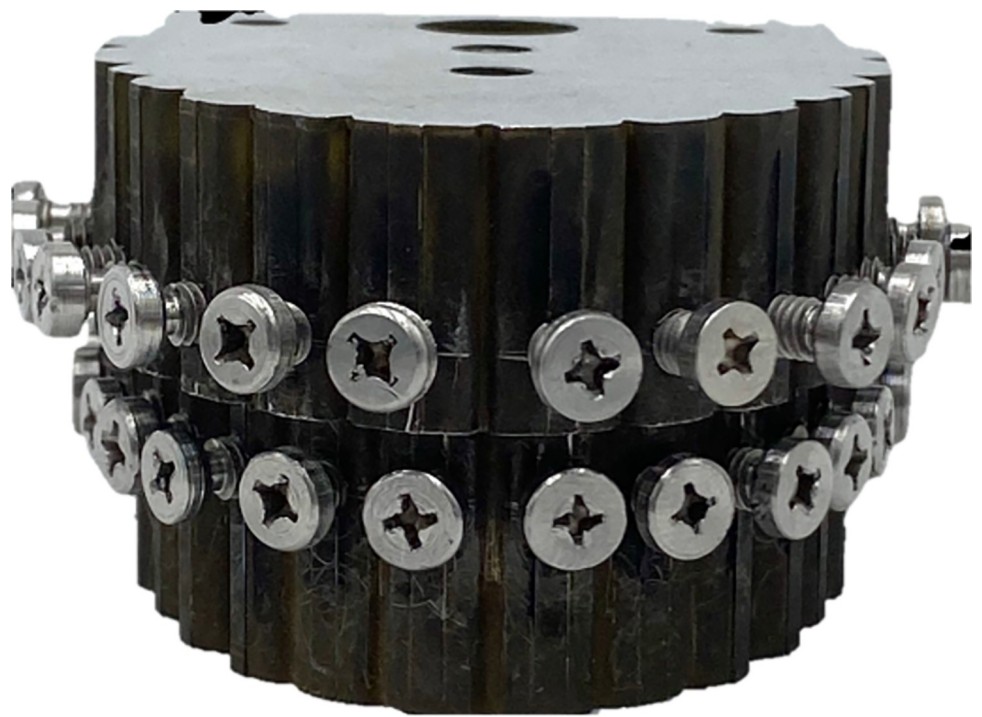

**Fig 1. Custom-made dental models through CAD/CAM technique.**

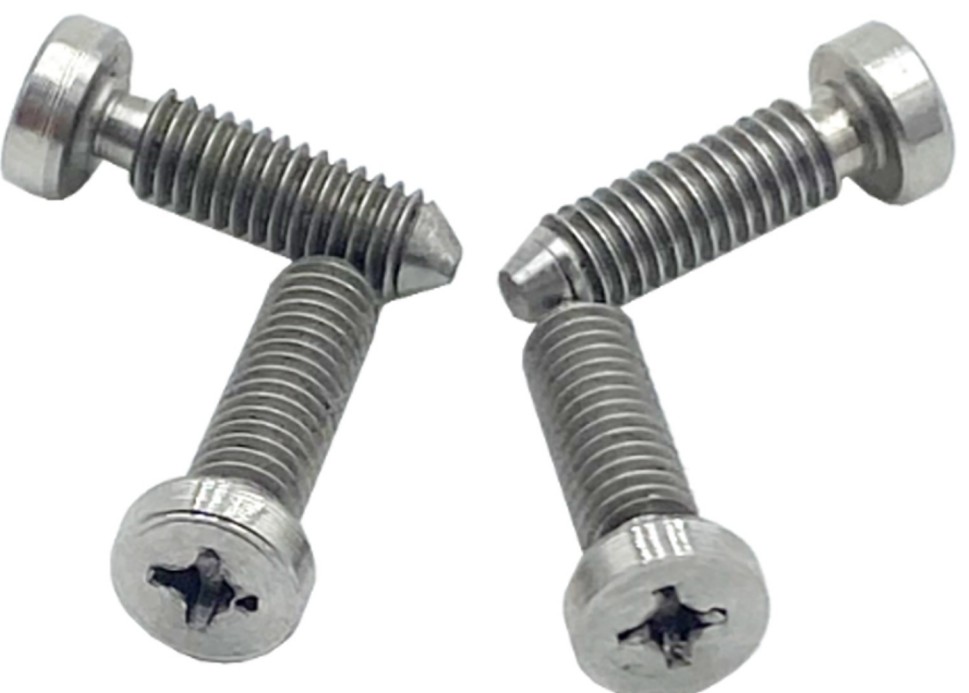

**Fig 2. Screws designed and fabricated to represent IMF screws and orthodontic brackets.**

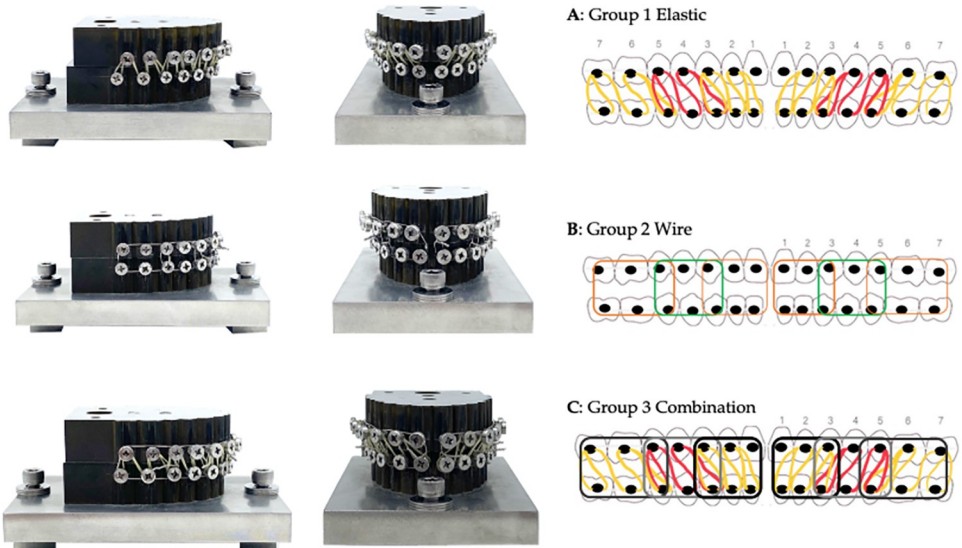

**Fig 3. (A, B, C).** Different MMF techniques representing all 3 sample groups.

were applied and were selected as the representative sample groups representing wire and elastic techniques in this study.

The study's design comprised of three sample groups consisting of wire (Fig 3B), elastic (Fig 3A), and a combination group (Fig 3C); which combined the fixation pattern of both the wire and elastic groups. Sample size calculation using Stata/SE 15.1 for power analysis for one-way analysis of variance with statistical significance set at 0.001 and a power of 0.99 was calculated using data from the pilot study. According to the calculation, a minimum sample group of 4 was established. However, for this particular study, a larger sample group of 10 was employed. The samples were tested using a 1000 Newton load cell using the Instron 5566 Universal Testing Machine (Instron Ltd., Buckinghamshire, England) (DT-40-73). Maxilloman-dibular fixation was executed in each sample according to the specified sample group parameters and loaded on and tested using the universal testing machine until 0.5 mm and 1 mm of vertical separation between the upper and lower models was achieved as demonstrated in Fig 4A–4C. Each process was repeated 10 times and the average peak tensile load was recorded and plotted.

Descriptive statistics was calculated using Microsoft Excel and SPSS (Statistical Package for the Social Sciences) version 25 (IBM, Chicago, IL, USA). Data was analyzed using a one-way ANOVA and multiple comparisons between each group was evaluated for statistical signifi-cance using Bonferroni's Test. The level of significance was set at $p < 0.05$. A comparison between all 3 groups is shown in Table 1.

## 3. Results

This study showed that the average tensile load to achieve 1 mm of vertical separation of the models in the elastic group was at 164.91±11.25 N, the wire group at 498.50±24.37 N, and the combination group at 741.91±34.90 N (Fig 5). Statistical analyses using one-way ANOVA showed the average tensile load difference between all 3 groups was statistically significant at the 0.05 level ($p < 0.001$) (Table 1, Fig 5).

Multiple comparisons was therefore performed using Bonferroni's test to evaluate the sta-tistical difference among the 3 groups. The mean difference in tensile load to resist 1 mm

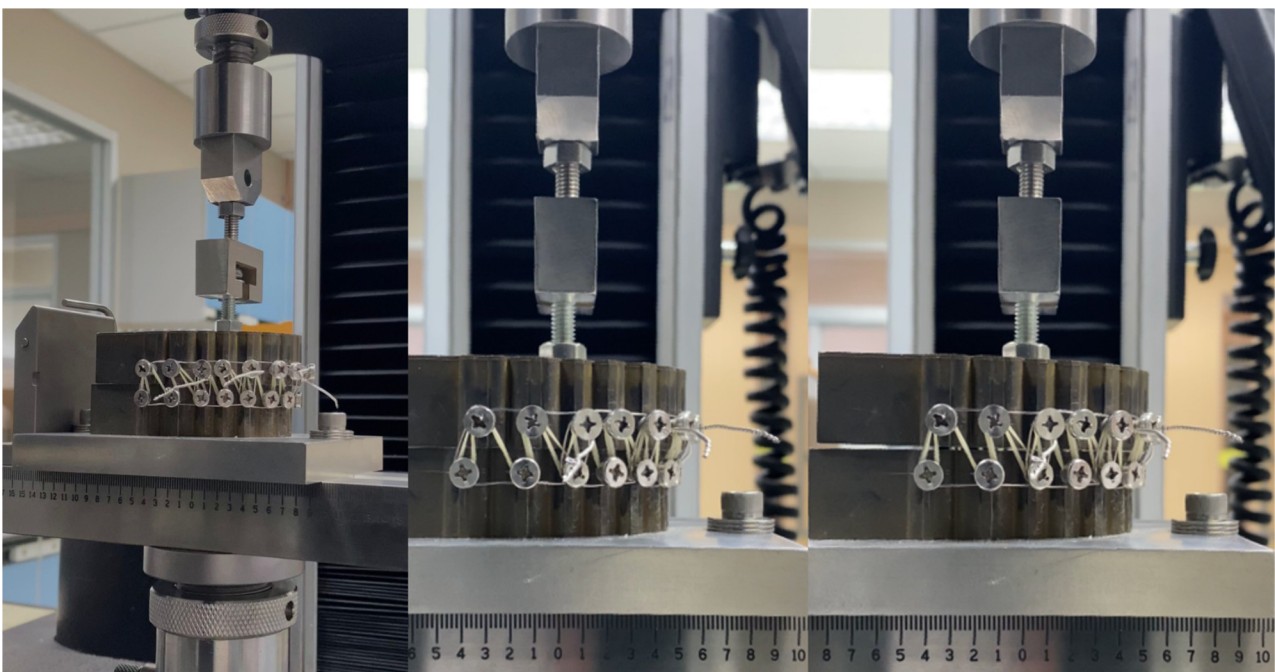

**Fig 4. A.** Orientation of dental models on the Instron 5566 Universal Testing Machine, **B.** Models at rest position at 0 mm separation, **C.** Models at complete 1 mm separation (complete cycle).

separation between the wire and elastic groups was statistically significant at the 0.05 level (p < .001). A mean difference in tensile load was 333.59 N (95%Cl 304.57, 362.61), exhibiting that the wire group had an average tensile load 3.02 times greater than that of the elastic group. When comparing the combination group to the elastic group, the statistical analysis also showed significance at the 0.05 level (p < .001). A mean difference in tensile load was at 576.99 N (95%Cl 547.98, 606.02) or 4.50 times greater than the elastic group (Fig 5). Lastly, when comparing the combination group to the wire group, statistical significance was also reported showing a mean difference in tensile load at 243.91 N (95%Cl 214.39, 272.43), showing the combination group had an average tensile load to be 1.49 times greater than the average load of the wire group (Table 2, Fig 5).

Statistical analyses of the same subject groups were also performed evaluating the separa-tion of the dental models at 0.5 mm using the same study parameters. The average tensile load in the elastic group was 125.59±8.22 N, the wire group at 308.36±18.67 N, and the combina-tion group at 487.45±31.98 N, respectively (Fig 6). Statistical analysis using one-way ANOVA showed the average tensile load difference between all 3 groups was statistically significant at 0.05 (p<0.001) (Table 1). Multiple comparisons were therefore performed using Bonferroni's test to evaluate the statistical difference between all 3 groups. The mean difference in tensile

**Table 1. Comparing tensile loads using elastic, wire, and combination methods.**

|  | Elastic | Wire | Combination | P-value |
|---|---|---|---|---|
| Load (N): 1 mm | 164.91±11.25 | 498.50±24.37 | 741.91±34.90 | <0.001* |
| Load (N): 0.5 mm | 125.59±8.22 | 308.36±18.67 | 487.45±31.98 | <0.001* |

Data were expressed as mean±SD and analyzed with One-way ANOVA

*Statistically significant at the 0.05 level

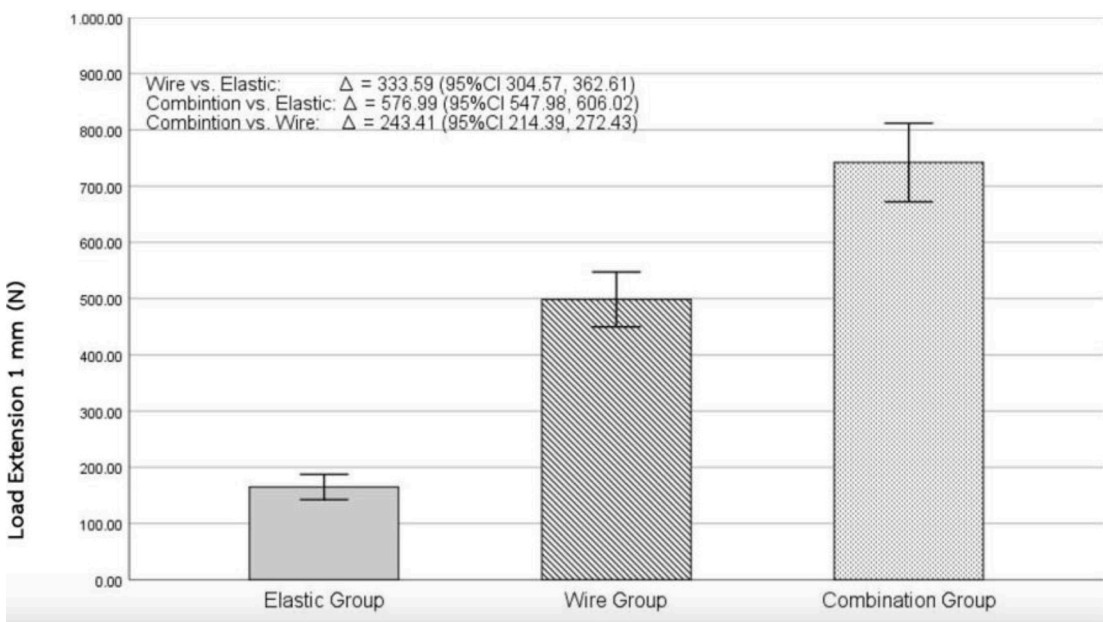

**Fig 5. Mean tensile loads of maxillo-mandibular fixation between all 3 groups at 1 mm separation.**

load to resist 0.5 mm separation between the wire group compared to the elastic was statistically significant at the 0.05 level (p<0.001). A mean difference in tensile load was 182.77 N (95%Cl 157.77, 207.76), displaying that the wire group had an average tensile load 2.46 times greater than that of the elastic group. When comparing the combination group to the elastic group, the statistical analysis also showed significance at the 0.05 level (p<0.001). A mean difference in tensile load was at 361.85 N (95%Cl 336.86, 386.85) or the combination group could resist displacing forces 3.88 times that of the elastic group. Lastly, when comparing the combination group to the wire group, statistical significance was also reported showing a mean difference in tensile load at 179.09 N (95%Cl 154.09, 204.08), meaning the combination group had an average tensile load to be 1.58 times the average load of the wire group (Table 2, Fig 6). The data from this study was therefore plotted as shown in Fig 7, showing a line graph that demonstrates the differences in tensile loads between all 3 sample groups at 0.5 and 1 mm vertical separation.

**Table 2. Multiple comparison between sample groups.**

| Group vs. Group | Mean difference (95%CI) | Standard error | P-value |
|---|---|---|---|
| **Load (N): Extension 1 mm** | | | |
| Wire vs. Elastic | 333.59 (304.57, 362.61) | 11.37 | <0.001* |
| Combination vs. Elastic | 576.99 (547.98, 606.02) | 11.37 | <0.001* |
| Combination vs. Wire | 243.41 (214.39, 272.43) | 11.37 | <0.001* |
| **Load: Extension 0.5 mm (N)** | | | |
| Wire vs. Elastic | 182.77 (157.77, 207.76) | 9.79 | <0.001* |
| Combination vs. Elastic | 361.85 (336.86, 386.85) | 9.79 | <0.001* |
| Combination vs. Wire | 179.09 (154.09, 204.08) | 9.79 | <0.001* |

Multiple comparisons were analyzed with Bonferroni method

* The mean difference is significant at the 0.05 level

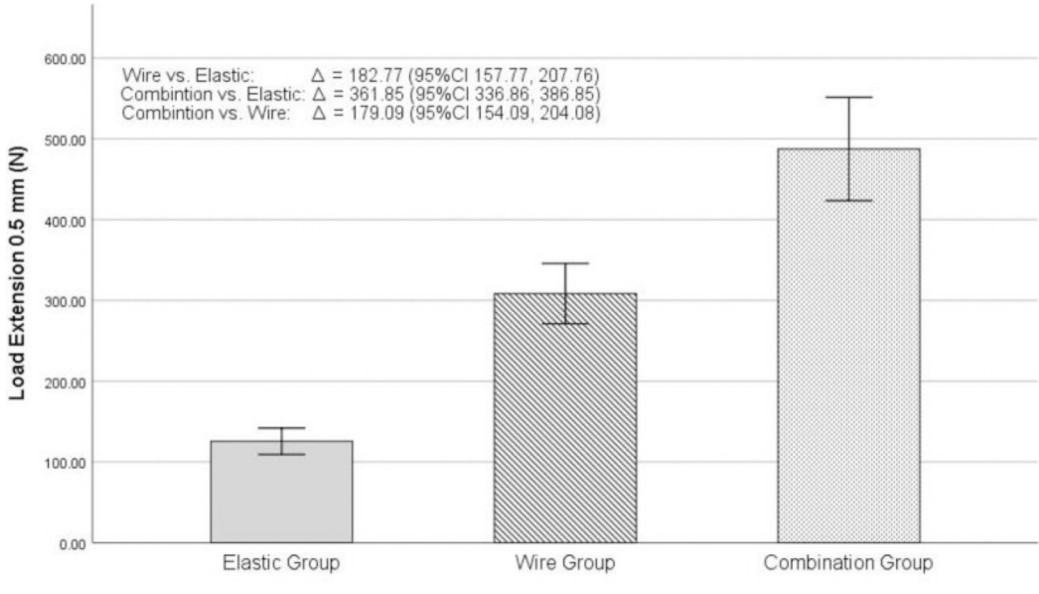

**Fig 6. Mean tensile loads of maxillo-mandibular fixation between at 3 groups at 0.5 mm separation.**

## 4. Discussion

In this conducted in vitro study, a comprehensive analysis was performed to assess the ability of three distinct maxillo-mandibular fixation techniques to withstand opposing forces. These techniques comprised of the elastic group, the wire group, and the combination group.

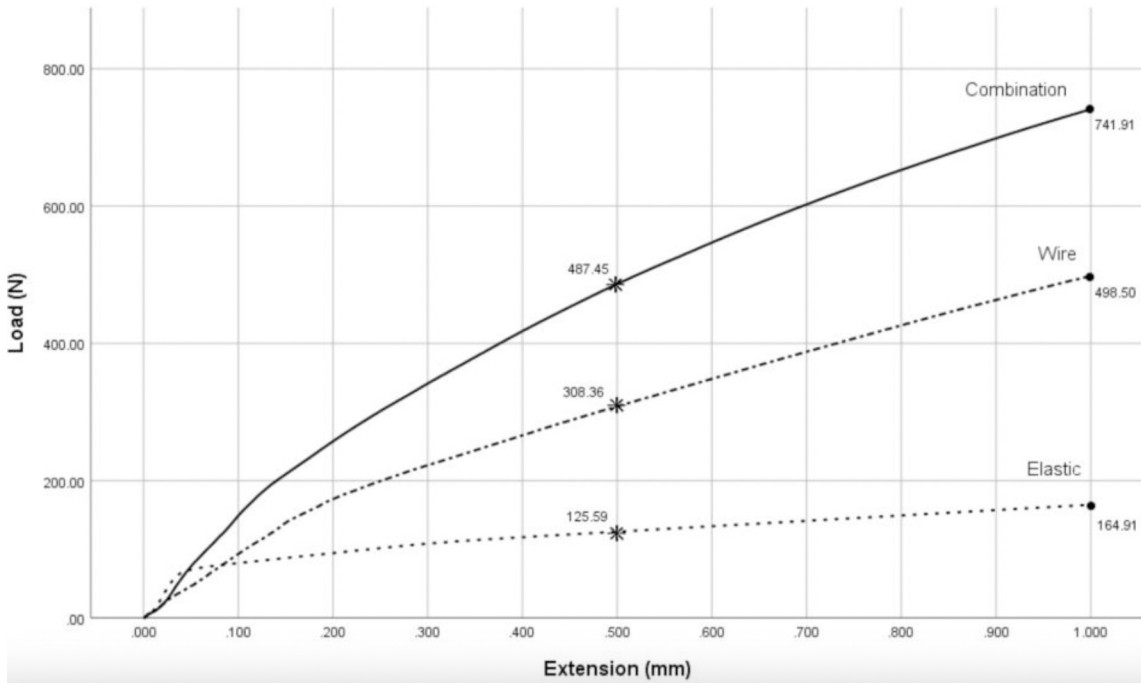

**Fig 7. Line graph showing the average tensile loads at both 0.5 and 1 mm separation.**

Statistical significance was observed in this comparison, yielding valuable insights into their relative efficacy. Data shows that the best fixation technique was the combination group followed by the wire group and lastly the elastic group, respectively. When analyzing the average tensile loads that each of these groups could endure before leading to a separation of at least 0.5 mm between the dental models, the elastic group emerged as the least resistant to opposing forces. The data further highlights a substantial discrepancy between the group with highest force-resisting capability (the combination group) and the group with the lowest (the elastic group). This disparity was significant, revealing a striking 400% difference in their respective abilities to withstand opposing forces. These findings substantiate the superiority of the combination technique in achieving robust maxillo-mandibular fixation, positioning it as the most effective among the evaluated techniques.

In 1993, A.T. Smith highlighted the potential risk of needlestick injuries associated with traditional wiring in contaminated environments, posing health hazards to healthcare providers. A systematic review by Bouya et al. in 2020 demonstrated the alarming global prevalence of needlestick injuries, some of which were attributed to intraoperative wire-related incidents, affecting as many as 2 million healthcare workers annually [10]. While elastics could mitigate the needlestick risk, our in vitro study data establishes that relying solely on elastics for maxillo-mandibular fixation is insufficient in resisting opposing forces when compared to the wire and combination techniques.

A study in 1970 by Yildirim studied the maximum opening and closing forces exerted by different skeletal types. The sample groups consisted of skeletal open and closed bites. The study found that there was no statistical difference between opening forces between the two groups. The average opening forces measured in the study were equivalent to 137–169 N [11]. Whereas a study by Brunton et al. in 2018 [12] with 149 patients aged 20–60 years with overall good and general health found that in a large sample of participants of a broad age range and a demographically diverse background, jaw opening forces were greater in males than females. Men had greater maximum opening forces with median values of 157.77 N compared to women at 166.61 N [12]. Our in vitro study data sheds light on maxillo-mandibular fixation using elastics alone. It demonstrates that elastics can only resist tensile loads causing a 0.5 mm separation of dental models at 126 N and a 1 mm separation at 165 N which is insufficient to resist the maximum opening forces in men and women as studied by Yilridum and Brunton et al. Transitioning to clinical scenarios, we reviewed the maximum push force that can be exerted through human force, particularly relevant during the surgical phase when applying temporary maxillo-mandibular fixation application or fracture reduction phases. Das et al. in 2004 examined isometric pull and push strength profiles in both genders and across different positions. Notably, the study found that maximum push strength in males ranged from 140 N to 227 N, depending on the seated or standing position. In females, the values ranged from 96 N to 140 N, again based on seating and standing positions [13].

This comprehensive examination of prior research and our in vitro findings collectively underscores the intricate interplay between technique efficacy, anatomical factors, and the practical forces encountered in clinical scenarios, providing a nuanced perspective on maxillo-mandibular fixation and its potential clinical implications. This study, conducted in a controlled in vitro environment utilizing the Instron 5566 Universal Testing Machine to vertically displace dental models, provides an initial understanding of tensile loads. However, it's vital to acknowledge that in actual anatomical scenarios, opposing forces between the mandible and maxilla encompass multidirectional dimensions, not merely vertical and horizontal. In light of these considerations, it is prudent to interpret the numerical findings from this study with a view toward more comprehensive anatomical evaluations. Alternatively, a clinical study might be envisaged to validate the clinical implications of the study's findings. In the future,

investigations should delve into the tensile strength exhibited when models are displaced in multiple directions, aiming to provide a more comprehensive and robust understanding of maxillo-mandibular fixation mechanics in complex clinical settings.

## 5. Conclusion

From this study, it can be established that maxillomandibular fixation using the combination technique could resist forces at least 1.5x or 150% more when compared to the wire group and at least 4x or 400% more when compared to the elastic group as shown in Fig 7.

Furthermore, a conclusion can be made after reviewing the maximum opening forces exerted by human force and maximum push force exerted in both a standing and sitting position that using elastic alone for maxillomandibular fixation cannot achieve adequate stability sufficient in opposing displacing forces during reduction and fixation. As this study is a pilot study and the first of its kind to study the tensile load from maxillo-mandibular fixation, additional studies may be needed before a clinical recommendation can be made.

## Supporting information

**S1 Fig. Appendix A.** Data collected from pilot study.
(PDF)

**S1 Data.**
(ZIP)

## Acknowledgments

The authors would like to acknowledge Mr. Thammasak Vimonkiattikun for his expertise and assistance in the designing and milling process of the customized dental models and screws.

## Author Contributions

**Conceptualization:** Sudeep Pawa, Surakit Visuttiwattanakorn.

**Data curation:** Sudeep Pawa.

**Formal analysis:** Sudeep Pawa.

**Investigation:** Sudeep Pawa.

**Methodology:** Sudeep Pawa.

**Supervision:** Surakit Visuttiwattanakorn.

**Writing – original draft:** Sudeep Pawa.

**Writing – review & editing:** Sudeep Pawa.

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
