## [Decision Letter · Decision Letter 0]

4 Jul 2023

PONE-D-23-14421An In Vitro Study in Separating Tensile Loads during Maxillo-Mandibular Fixation using Wire and/or ElasticsPLOS ONE

Dear Dr. Visuttiwattanakorn,

Thank you for submitting your manuscript to PLOS ONE. After careful consideration, we feel that it has merit but does not fully meet PLOS ONE’s publication criteria as it currently stands. Therefore, we invite you to submit a revised version of the manuscript that addresses the points raised during the review process.

Please address the comments from the reviewers and editor. Please pay attention to the following comments.

1) Abstract: Condense the introduction in the abstract and focus more on the meaning of the findings and their clinical implications.

2) General format: Ensure that the manuscript follows the PLOS ONE format by consulting the information for authors.

3) Introduction: Improve the introduction by clearly stating the research problem or objective, establishing the need for further investigation, discussing the advantages and limitations of different fixation methods, and restructuring the content for better organization and flow.

4) Methods: Add information on how the model simulates clinical situations or discuss the application of the results to clinical patients if no clinical validation is available.

5) Discussion and Conclusion: Separate the Discussion and Conclusion sections, following a format that highlights important results, compares them with previous studies, discusses study limitations, and indicates clinical implications and future research. Move Figure 7 to the Results section.

6) Conclusion: Provide a clear summary of the results in terms of addressing the hypothesis, considering the manuscript's lack of a clear clinical problem and hypothesis.

We look forward to receiving your revised manuscript.

Kind regards,

Sompop Bencharit, DDS, MS, PhD, FACP

Academic Editor

PLOS ONE

3. We note that Figures 1, 2, 3, and 4 in your submission contain copyrighted images. All PLOS content is published under the Creative Commons Attribution License (CC BY 4.0), which means that the manuscript, images, and Supporting Information files will be freely available online, and any third party is permitted to access, download, copy, distribute, and use these materials in any way, even commercially, with proper attribution. For more information, see our copyright guidelines: http://journals.plos.org/plosone/s/licenses-and-copyright.

1. You may seek permission from the original copyright holder of Figures 1, 2, 3, and 4 to publish the content specifically under the CC BY 4.0 license.

Additional Editor Comments:

Thank you for submitting your work to PLOS ONE. The reviewers have provided a range of feedback, including suggestions for minor revision and acceptance. In addition to addressing the reviewers' comments, I request that you make the following revisions to the manuscript:

1) Abstract: Condense the first three sentences into one to provide a concise overview. Focus more on explaining the meaning of the findings and their clinical implications.

2) General format: Ensure that the manuscript follows the PLOS ONE format. Please consult the information for authors to ensure compliance.

3) Introduction: Although the introduction provides a comprehensive overview of intermaxillary fixation (IMF) or maxillo-mandibular fixation (MMF), there are areas for improvement:

3.1) Clearly state the research problem or objective in a concise manner. Highlight the specific research question or knowledge gap that the study aims to address.

3.2) Establish the need for further investigation or improvement by discussing the limitations, challenges, or evidence gaps associated with current techniques. Justify the need for the study by providing relevant information from the literature.

3.3) Elaborate on the advantages and limitations of orthodontic latex elastics as an alternative to wire ligatures. Briefly discuss the advantages and disadvantages of different fixation methods to enhance the reader's understanding of the research context.

3.4) Restructure the introduction to ensure a logical flow and organization of information. Improve transitions between ideas to present a coherent and logical progression of concepts.

4) Methods: While the methods provide sufficient experimental details, include information on how the model simulates clinical situations. If there is clinical validation, provide relevant details. If not, discuss how the results can be applied to clinical patients.

5) Discussion and Conclusion: Separate the Discussion and Conclusion sections. The Discussion should follow this format: 4.1) highlight important results, 4.2) compare the results with previous studies, 4.3) discuss the limitations of the study, and 4.4) indicate clinical implications and suggest future studies. Move Figure 7 from the Discussion section to the Results section.

6) Conclusion: Provide a clear and direct summary of the results in terms of addressing the hypothesis. Since the manuscript lacks a clear clinical problem and hypothesis, ensure the conclusion is based on the obtained results.

Reviewers' comments:

Reviewer's Responses to Questions

**Comments to the Author**

1. Is the manuscript technically sound, and do the data support the conclusions?

Reviewer #1: Partly

Reviewer #2: Yes

Reviewer #3: Yes

2. Has the statistical analysis been performed appropriately and rigorously? 

Reviewer #1: I Don't Know

Reviewer #2: Yes

Reviewer #3: I Don't Know

3. Have the authors made all data underlying the findings in their manuscript fully available?

Reviewer #1: No

Reviewer #2: Yes

Reviewer #3: Yes

4. Is the manuscript presented in an intelligible fashion and written in standard English?

Reviewer #1: No

Reviewer #2: Yes

Reviewer #3: Yes

5. Review Comments to the Author

Reviewer #1: To begin with

English language editing services very much needed for manuscript.

This is a CAD based design which was utilised for this invitro study . Wat was the printing material utilised for this study????

Please provide justifications and citations to say the model material doesn’t interferes with testing results.

Please justify your sample size

Please add more correct citations under discussion. Total references of 13 for an original article is unacceptable.

Wat is new in this study when compared to previously published articles on similar topic.

Please provide relevant citations to your materials and methods section.

Your study is on tensile load only. Wat about shear loads acting on mandible at time of Fixation. Anything included in your study ???

Reviewer #2: Nice preclinical project, not without its limitations as noted. Well written and documented. A couple of minor suggestions:

1. although the outcomes will not change, I would recommend indicating the parameters that were tested and to what level they were tested when completing the power analysis.

2. Not being an OMS or orthodontist and being familiar with the up to date state of MMF and arrangement of wires and elastics commonly used in MMF, it might help the reader to know what currently is the standard configuration that is taught and used in clinical practice to achieve fixation. If internal fixation is used is external fixation even needed? The OMS department in my current institution rarely in orthognatic cases uses external fixation once the internal fixation is placed. I would make comment of the current gold standard, and in what cases would you prefer external vs internal fixation. Additionally, I would also maybe make more comment of the consequences of inadequate fixation in the introduction.

3. Discuss experimental model limitations in greater detail and contrasting to what is done clinically- while MM bone fixation screws may be used (I am guessing not usually one for every tooth as used in this model), as indicated orthodontic brackets +- MMF screws are more commonly used. Why was one screw used for every tooth? What was the rationale? Why were orthodontic brackets not used if this is more commonly used in clinical practice? If repeating or extending this experimental model to study the effects of lateral forces, I would suggest integrating orthodontic brackets into the model (if you are able to find a way to bond to the custom 'dental model') to better represent what may be done clinically. This may be a point for discussion in the limitations of this preclinical model, along with discussion of any potential differences in the size of the custom fixation screw that was used in this study, to what it commonly available on the market (they seem quite large in diameter and not what may be used clinically). Granted this is a preclinical study with obvious limitations, clinical correlation and deeper discussion on model limitations is helpful for the reader as if products/screws that are used are not commercially available or the set up is not as it would be in a real situation then the results have minimal clinical meaning.

Other than that, nice project. I would be interested to see as you mentioned how it would fare up with lateral movements which if the patient is a bruxer may be the time when potentially higher forces are generated and this could impact the fixation.

Reviewer #3: The study is well designed and the manuscript is also acceptable for publication. However, it would be better if the clinical implication(s) of the in vitro findings are highlighted. The article seems to lack novelty.

6. PLOS authors have the option to publish the peer review history of their article (what does this mean?). If published, this will include your full peer review and any attached files.

Reviewer #1: **Yes: **Prof Dr. Hariharan Ramakrishnan

Reviewer #2: No

Reviewer #3: No

---

## [Author Response · Author response to Decision Letter 0]

3 Sep 2023

Dear Editor and Reviewers,

 On behalf of all authors, we are grateful for your helpful in-depth feedback on our manuscript. We hope that you consider all raised points to be adequately addressed by our revised manuscript as described below and that the revised manuscript qualifies for publication.

Kind Regards,

Surakit Visutiwattanakorn

Editor Comments

1. Abstract: Condense the first three sentences into one to provide a concise overview. Focus more on explaining the meaning of the findings and their clinical implications.

The abstract has been revised to provide a more concise overview of the manuscript.

2. General format: Ensure that the manuscript follows the PLOS ONE format. Please consult the information for authors to ensure compliance.

The manuscript has been edited to follow the PLOS ONE format as written on the “information for authors” page.

3. Introduction: Although the introduction provides a comprehensive overview of intermaxillary fixation (IMF) or maxillo-mandibular fixation (MMF), there are areas for improvement:

3.1) Clearly state the research problem or objective in a concise manner. Highlight the specific research question or knowledge gap that the study aims to address.

3.2) Establish the need for further investigation or improvement by discussing the limitations, challenges, or evidence gaps associated with current techniques. Justify the need for the study by providing relevant information from the literature.

3.3) Elaborate on the advantages and limitations of orthodontic latex elastics as an alternative to wire ligatures. Briefly discuss the advantages and disadvantages of different fixation methods to enhance the reader's understanding of the research context.

3.4) Restructure the introduction to ensure a logical flow and organization of information. Improve transitions between ideas to present a coherent and logical progression of concepts.

The introduction has been rewritten and involves all the changes recommended by the editor and reviewers. The research problem and objectives have been clearly stated so that the reader clearly understands the knowledge gap and what this study aims to address.

4. Methods: While the methods provide sufficient experimental details, include information on how the model simulates clinical situations. If there is clinical validation, provide relevant details. If not, discuss how the results can be applied to clinical patients.

The materials and methods have been reformatted to better explain how the custom-made models simulate real-life scenarios. The purpose of the study has been stated to further discuss how the results can be applied.

5. Discussion and Conclusion: Separate the Discussion and Conclusion sections. The Discussion should follow this format: 4.1) highlight important results, 4.2) compare the results with previous studies, 4.3) discuss the limitations of the study, and 4.4) indicate clinical implications and suggest future studies. Move Figure 7 from the Discussion section to the Results section.

The discussion and conclusion sections have been separated as recommended. Figure 7 has been moved and re-cited within the results section.

6. Conclusion: Provide a clear and direct summary of the results in terms of addressing the hypothesis. Since the manuscript lacks a clear clinical problem and hypothesis, ensure the conclusion is based on the obtained results.

The conclusion has been totally re-written as per the editors and reviewer’s recommendation.

Journal Requirements

I have re-formatted my manuscript according the PLOS ONE’s style requirements. Please let me know if any additional changes need to be made.

2. PLOS requires an ORCID iD for the corresponding author in Editorial Manager on papers submitted after December 6th, 2016. Please ensure that you have an ORCID iD and that it is validated in Editorial Manager.

I have updated my profile to include my ORCID iD in the Editorial Manager Portal.

3. We note that Figures 1, 2, 3, and 4 in your submission contain copyrighted images. All PLOS content is published under the Creative Commons Attribution License (CC BY 4.0), which means that the manuscript, images, and Supporting Information files will be freely available online, and any third party is permitted to access, download, copy, distribute, and use these materials in any way, even commercially, with proper attribution.

Figures 1-4 are original photographs taken by Sudeep Pawa during the experimentation phase of this study. Is it still necessary to have the copyright paperwork for this situation?

Reviewer #1: To begin with

- English language editing services very much needed for manuscript.

- This is a CAD based design which was utilised for this invitro study . Wat was the printing material utilised for this study????

- Please provide justifications and citations to say the model material doesn’t interferes with testing results.

- Please justify your sample size

- Please add more correct citations under discussion. Total references of 13 for an original article is unacceptable.

- Wat is new in this study when compared to previously published articles on similar topic.

- Please provide relevant citations to your materials and methods section.

- Your study is on tensile load only. Wat about shear loads acting on mandible at time of Fixation. Anything included in your study ???

Reply to Reviewer # 1

We sincerely appreciate your insightful and thorough review of our manuscript. The manuscript has undergone grammatical revisions. As detailed within, the customized models were created through the utilization of Computer-Aided Design (CAD) and subsequently fabricated via Computer-Aided Manufacturing (CAM). These models were constructed utilizing Stainless Steel as the chosen material. While material selection lacks specific justification, it is worth noting that material properties did not influence this study's scope, which solely focused on analyzing Tensile Loads causing separation between the two models.

In the context of sample size, considering our status as a pilot study embarking on the pioneering examination of tensile loads in Maxillo-Mandibular Fixation, an initial pilot study was conducted. Sample size determination was carried out employing Stata/SE 15.1 for power analysis concerning one-way analysis of variance. This calculation incorporated a statistical significance level of 0.001, alongside a power value of 0.99, drawing on insights garnered from the pilot study's data.

Given the absence of prior experimental investigations into tensile loads encompassing varied patterns during maxillo-mandibular fixation, it is important to acknowledge that direct comparisons with previous research cannot be made. Furthermore, with regards to shear loads impacting the mandible, the study was intentionally designed to explore vertical (unidirectional) forces exclusively. It is acknowledged that this design choice represents a limitation within our study.

Reviewer #2: Nice preclinical project, not without its limitations as noted. Well written and documented. A couple of minor suggestions:

1. although the outcomes will not change, I would recommend indicating the parameters that were tested and to what level they were tested when completing the power analysis.

2. Not being an OMS or orthodontist and being familiar with the up to date state of MMF and arrangement of wires and elastics commonly used in MMF, it might help the reader to know what currently is the standard configuration that is taught and used in clinical practice to achieve fixation. If internal fixation is used is external fixation even needed? The OMS department in my current institution rarely in orthognatic cases uses external fixation once the internal fixation is placed. I would make comment of the current gold standard, and in what cases would you prefer external vs internal fixation. Additionally, I would also maybe make more comment of the consequences of inadequate fixation in the introduction.

3. Discuss experimental model limitations in greater detail and contrasting to what is done clinically- while MM bone fixation screws may be used (I am guessing not usually one for every tooth as used in this model), as indicated orthodontic brackets +- MMF screws are more commonly used. Why was one screw used for every tooth? What was the rationale? Why were orthodontic brackets not used if this is more commonly used in clinical practice? If repeating or extending this experimental model to study the effects of lateral forces, I would suggest integrating orthodontic brackets into the model (if you are able to find a way to bond to the custom 'dental model') to better represent what may be done clinically. This may be a point for discussion in the limitations of this preclinical model, along with discussion of any potential differences in the size of the custom fixation screw that was used in this study, to what it commonly available on the market (they seem quite large in diameter and not what may be used clinically). Granted this is a preclinical study with obvious limitations, clinical correlation and deeper discussion on model limitations is helpful for the reader as if products/screws that are used are not commercially available or the set up is not as it would be in a real situation then the results have minimal clinical meaning.

Other than that, nice project. I would be interested to see as you mentioned how it would fare up with lateral movements which if the patient is a bruxer may be the time when potentially higher forces are generated and this could impact the fixation.

Reply to Reviewer # 2

We sincerely appreciate your insightful and thorough review of our manuscript. The study has taken steps to enhance reader comprehension by rephrasing the study's parameters and detailing confidence levels or p-values. Regarding the established benchmark for achieving satisfactory Maxillomandibular Fixation (MMF), it's important to note that both techniques continue to be employed globally without a definitive consensus on a superior method. Concerning internal and external fixation, External Fixation (MMF) is frequently employed to facilitate precise and anatomical internal fixation. In the postoperative phase, external fixation serves to guide occlusion training in numerous instances. The introduction has now been supplemented with information on the ramifications of insufficient MMF.

For the purpose of this pilot study and to mitigate variables influencing outcomes, the authors opted for the use of screws instead of orthodontic brackets. This strategic choice eliminated potential factors tied to bracket bond strength, which would have been pertinent had brackets been employed. Clear delineations have been provided about the study's limitations, along with suggestions for future research based on this study's findings

Reviewer #3: The study is well designed and the manuscript is also acceptable for publication. However, it would be better if the clinical implication(s) of the in vitro findings are highlighted. The article seems to lack novelty.

Reply to Reviewer # 3

We sincerely appreciate your insightful and thorough review of our manuscript. Clinical implications of this in vitro study have been added. We hope that the changes are sufficient to qualify for publication.

---

## [Decision Letter · Decision Letter 1]

17 Oct 2023

PONE-D-23-14421R1An In Vitro Study in Separating Tensile Loads during Maxillo-Mandibular Fixation using Wire and/or ElasticsPLOS ONE

Dear Dr. Visuttiwattanakorn,

Thank you for submitting your manuscript to PLOS ONE. After careful consideration, we feel that it has merit but does not fully meet PLOS ONE’s publication criteria as it currently stands. Therefore, we invite you to submit a revised version of the manuscript that addresses the points raised during the review process.

While the reviewers felt that the revised manuscript is much improved, there are some minor issues as well as the study rationale needed to be addressed. Please address these comments thoroughly. 

We look forward to receiving your revised manuscript.

Kind regards,

Sompop Bencharit, DDS, MS, PhD, FACP

Academic Editor

PLOS ONE

Additional Editor Comments:

While the reviewers felt that the revised manuscript is much improved, there are some minor issues as well as the study rationale needed to be addressed. Please address these comments thoroughly.

Reviewers' comments:

Reviewer's Responses to Questions

**Comments to the Author**

1. If the authors have adequately addressed your comments raised in a previous round of review and you feel that this manuscript is now acceptable for publication, you may indicate that here to bypass the “Comments to the Author” section, enter your conflict of interest statement in the “Confidential to Editor” section, and submit your "Accept" recommendation.

Reviewer #1: (No Response)

Reviewer #2: All comments have been addressed

Reviewer #3: (No Response)

2. Is the manuscript technically sound, and do the data support the conclusions?

Reviewer #1: Yes

Reviewer #2: Yes

Reviewer #3: Partly

3. Has the statistical analysis been performed appropriately and rigorously? 

Reviewer #1: I Don't Know

Reviewer #2: Yes

Reviewer #3: I Don't Know

4. Have the authors made all data underlying the findings in their manuscript fully available?

Reviewer #1: No

Reviewer #2: Yes

Reviewer #3: Yes

5. Is the manuscript presented in an intelligible fashion and written in standard English?

Reviewer #1: Yes

Reviewer #2: Yes

Reviewer #3: Yes

6. Review Comments to the Author

Reviewer #1: I still see Abstract is not a structured one . Please provide a structured abstract. Please provide all data related to this study in a repository or a supplemental file . I meant all those files which had not been mentioned in the revised manuscript.

Reviewer #2: - I question the use, discussion and relevance of reference #13 on line 234. It appears the study referenced is in relation to a persons ability to pull down on a level with their arm in a seating and standing position. I fail to see how this study is relevant to the discussion of forces that could potentially be applied in the oral region. I am sure there are more appropriate references to discuss, allowing for a clinical correlation/relevance. This may also enlighten us as to if doing the combined fixation is the way to go, or is the additional stability achieved in this model irrelevant and so standard wire fixation is all that is needed?

- another clinical relevance note - forces from the elastics degrade over time and so if this type of fixation is needed for some time, elastics will need to be replaced.

- I would move the discussion on model limitations to the end of the discussion section and not in the conclusions section

Reviewer #3: Dear doctor, despite your efforts, your experimental model has several inadequacies, that make the results invalid for clinical interpretation. The experimental model is merely indicating the tensile properties of various materials and is not even a faint replication of the clinical situation. The results obtained from this study are insignificant as long as they do not have any strong correlation with the clinical situation. Combining two materials will always give the concluded result. Please provide a strong rationale for your study, to justify its publication.

7. PLOS authors have the option to publish the peer review history of their article (what does this mean?). If published, this will include your full peer review and any attached files.

Reviewer #1: **Yes: **Prof Dr Hariharan Ramakrishnan

Reviewer #2: No

Reviewer #3: No

---

## [Author Response · Author response to Decision Letter 1]

19 Nov 2023

Rebuttal Letter

November 20th 2023,

Dear Editor and Reviewers,

On behalf of all authors, we are grateful for your helpful in-depth feedback on our manuscript. We hope that you consider all raised points to be adequately addressed by our revised manuscript as described below and that the revised manuscript qualifies for publication.

Kind Regards,

Surakit Visutiwattanakorn

Reviewer #1: I still see Abstract is not a structured one . Please provide a structured abstract. Please provide all data related to this study in a repository or a supplemental file . I meant all those files which had not been mentioned in the revised manuscript.

Reply to Reviewer #1: Thank you for your kind acknowledgment of our comprehensive review of your manuscript. We are glad to inform you that the abstract has been meticulously re-written and restructured to enhance clarity, facilitating a more effective understanding of the study's purpose and results.Additionally, we have included files that contain data from the Instron 5566 Universal Testing Machine ensuring transparency and completeness, in this research.

Reviewer #2: I question the use, discussion and relevance of reference #13 on line 234. It appears the study referenced is in relation to a persons ability to pull down on a level with their arm in a seating and standing position. I fail to see how this study is relevant to the discussion of forces that could potentially be applied in the oral region. I am sure there are more appropriate references to discuss, allowing for a clinical correlation/relevance. This may also enlighten us as to if doing the combined fixation is the way to go, or is the additional stability achieved in this model irrelevant and so standard wire fixation is all that is needed?

- another clinical relevance note - forces from the elastics degrade over time and so if this type of fixation is needed for some time, elastics will need to be replaced.

- I would move the discussion on model limitations to the end of the discussion section and not in the conclusions section

Reply to Reviewer #2: We truly value your thorough examination of our manuscript. Reference #13 was carefully selected as a benchmark, in our study due to its relevance in assessing the forces applied to the maxillo unit during intraoperative reduction and fixation. Our main goal was to present the findings from our in vitro experiments regarding the forces needed to cause separation of models during fixation. It is essential to clarify that our investigation focused on the intraoperative phase, examining the immediate effects of intermaxillary fixation, rather than addressing potential degradation of elastics over time, which may be a significant factor in the post-operative period. Our aim was to offer a numerical representation derived from in vitro experiments, intending to establish a reference point for future clinical studies. The discussion on model limitations has also been moved to the discussion section.

Reviewer #3: Dear doctor, despite your efforts, your experimental model has several inadequacies, that make the results invalid for clinical interpretation. The experimental model is merely indicating the tensile properties of various materials and is not even a faint replication of the clinical situation. The results obtained from this study are insignificant as long as they do not have any strong correlation with the clinical situation. Combining two materials will always give the concluded result. Please provide a strong rationale for your study, to justify its publication.

Reply to Reviewer #3: Given the absence of prior experimental investigations into tensile loads encompassing varied patterns during maxillo-mandibular fixation, it is important to acknowledge that direct comparisons with previous research cannot be made. Furthermore, with regards to shear loads impacting the mandible, the study was intentionally designed to explore vertical (unidirectional) forces exclusively. It is acknowledged that this design choice represents a limitation within our study. For the purpose of this pilot study and to mitigate variables influencing outcomes, the authors opted for the use of screws instead of orthodontic brackets. This strategic choice eliminated potential factors tied to bracket bond strength, which would have been pertinent had brackets been employed. Clear delineations have been provided about the study's limitations, along with suggestions for future research based on this study's findings. Our goal was to provide an in vitro numerical representative value that would serve as a reference for future clinical studies.

---

## [Decision Letter · Decision Letter 2]

27 Feb 2024

An In Vitro Study in Separating Tensile Loads during Maxillo-Mandibular Fixation using Wire and/or Elastics

PONE-D-23-14421R2

Dear Dr. Visuttiwattanakorn,

We’re pleased to inform you that your manuscript has been judged scientifically suitable for publication and will be formally accepted for publication once it meets all outstanding technical requirements.

Kind regards,

Sompop Bencharit, DDS, MS, PhD, FACP

Academic Editor

PLOS ONE

Additional Editor Comments (optional):

Thank you for your thorough responses to the reviewer.

Reviewers' comments:

Reviewer's Responses to Questions

**Comments to the Author**

1. If the authors have adequately addressed your comments raised in a previous round of review and you feel that this manuscript is now acceptable for publication, you may indicate that here to bypass the “Comments to the Author” section, enter your conflict of interest statement in the “Confidential to Editor” section, and submit your "Accept" recommendation.

Reviewer #1: (No Response)

Reviewer #2: All comments have been addressed

Reviewer #3: All comments have been addressed

2. Is the manuscript technically sound, and do the data support the conclusions?

Reviewer #1: Partly

Reviewer #2: Yes

Reviewer #3: (No Response)

3. Has the statistical analysis been performed appropriately and rigorously? 

Reviewer #1: I Don't Know

Reviewer #2: Yes

Reviewer #3: (No Response)

4. Have the authors made all data underlying the findings in their manuscript fully available?

Reviewer #1: Yes

Reviewer #2: Yes

Reviewer #3: (No Response)

5. Is the manuscript presented in an intelligible fashion and written in standard English?

Reviewer #1: No

Reviewer #2: Yes

Reviewer #3: (No Response)

6. Review Comments to the Author

Reviewer #1: (No Response)

Reviewer #2: (No Response)

Reviewer #3: The authors have addressed the shortcomings in the prior submission satisfactorily. They have acknowledged the limitations of the study. Clinicians must exercise caution while interpreting the concluded results.

7. PLOS authors have the option to publish the peer review history of their article (what does this mean?). If published, this will include your full peer review and any attached files.

Reviewer #1: **Yes: **Prof Dr Hariharan Ramakrishnan

Reviewer #2: No

Reviewer #3: No

---

## [Editor Report · Acceptance letter]

5 Mar 2024

PONE-D-23-14421R2 

PLOS ONE

Dear Dr. Visuttiwattanakorn, 

I'm pleased to inform you that your manuscript has been deemed suitable for publication in PLOS ONE. Congratulations! Your manuscript is now being handed over to our production team.

Kind regards, 

on behalf of

Dr. PLOS Manuscript Reassignment 

Staff Editor

PLOS ONE